# *PAX2* Gene Mutation in Pediatric Renal Disorders—A Narrative Review

**DOI:** 10.3390/ijms241612737

**Published:** 2023-08-13

**Authors:** Carmen Muntean, Camelia Chirtes, Balazs Baczoni, Claudia Banescu

**Affiliations:** 1Department of Pediatrics I, George Emil Palade University of Medicine, Pharmacy, Science and Technology of Targu Mures, 540142 Targu Mures, Romania; duicucarmen@yahoo.com; 2Laboratory of Genetics, Department of Genetics, Emergency County Hospital, 540142 Targu Mures, Romania; cameliamariaoprea@yahoo.com (C.C.); balazs.baczoni@gmail.com (B.B.); 3Center for Advanced Medical and Pharmaceutical Research, George Emil Palade University of Medicine, Pharmacy, Science and Technology of Targu Mures, 540142 Targu Mures, Romania

**Keywords:** *PAX2* gene, renal disorder, congenital anomalies, kidney, urinary tract, children

## Abstract

The *PAX2* gene is a transcription factor that is essential for the development of the urinary system among other transcription factors. The role of *PAX2* is highlighted from the seventh week of gestation, when it is involved in development processes and the emergence of nephrons and collecting tubes. Being an important factor in renal development, mutations of this gene can produce severe alterations in the development of the urinary tract, namely congenital anomalies of the kidneys and urinary tract. The first reported cases described with the *PAX2* mutation included both renal anomalies and the involvement of other organs, such as the eyes, producing renal coloboma syndrome. Over the years, numerous cases have been reported, including those with only renal and urinary tract anomalies. The aim of this review is to present a summary of pediatric patients described to have mutations in the *PAX2* gene to contribute to a better understanding of the genetic mechanism causing anomalies of the kidneys and urinary tract. In this review, we have included only pediatric cases with renal and urinary tract disorders, without the involvement of other organs. From what we know so far from the literature, this is the first review gathering pediatric patients presenting the PAX2 mutation who have been diagnosed exclusively with renal and urinary tract disorders.

## 1. Introduction

The *PAX2* (paired box gene 2) gene (MIM #167409), localized on chromosome 10q24.31, is a member of the family of PAX transcription factor genes, and it has a pivotal role in human urinary system development [1]. The *PAX* gene family is composed of four subgroups based on their structural similarities and also the presence of octapeptide, which are divided as follows: subgroup I contains PAX1 and PAX9; subgroup II (PAX2, PAX5 and PAX8 containing the paired box and octapeptide); subgroup III (PAX3 and PAX7); and subgroup IV containing the PAX4 and PAX6 genes as well as the paired box [2]. The *PAX2* gene (also known as *FSGS7*, *PAPRS*, *PAX-2*) consists of 14 coding exons and 13 introns.

Several transcription factors have proven to be essential in kidney embryogenesis, and, among these, *PAX2* also has a role in nephron development from the beginning of nephrogenesis [3]. *PAX2* is a homeotic gene that encodes an amino-terminal protein domain; this domain has been noted in proteins implicated in RNA polymerase II transcription involved in embryogenesis. It also can bind DNA and requires an octapeptide or homeodomain to function properly [2]. During the embryonic period, kidney development starts in week four with the nephrogenic cord formation and ends in week thirty-two when the nephrogenesis is complete and the kidney is functioning [3,4]. The *PAX2* gene is involved in the definitive kidney formation, which is present in the caudal intermediate mesoderm and is also expressed in the ureteric bud (UB) and the metanephric mesenchyme (MM) [1] (Figure 1).

In this process, *PAX2* recruits a histone H3K4 methyltransferase PAX interacting protein 1–mixed-lineage leukemia (PTIP–MLL) complex as a response to inductive signals and is associated with this complex by attaching to PTIP, which is part of the MLL3/4(KMT2C/D) histone H3K4 methyltransferase complex and connects PAX proteins to the H3K4me complex [5,6].

PTIP connects PAX proteins to MLL complexes for the purpose of sustaining gene activation, while the regulated co-repressor Groucho-related protein 4 (Grg4) has the ability to override this process to imprint suppressive epigenetic marks. As a result of interactions with histone-modulating complexes, it is thought that PAX proteins confer the ability to imprint both activating and repressive epigenetic marks, dividing the genome into active and silent regions in a cell-type-specific manner [7].

The transcription factors PAX2 and PAX8 play a crucial role in the embryonic development of the urogenital tract, especially in terms of nephron differentiation, branching formation and metanephric kidney induction [8,9].

**Figure 1 ijms-24-12737-f001:**
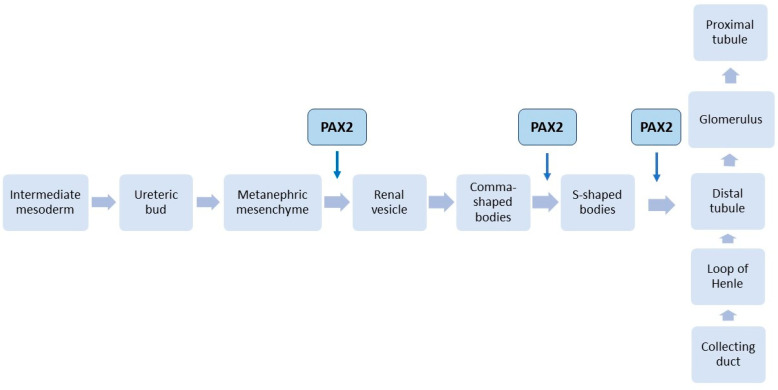
The ureteric bud appears in the seventh week of fetal development, arising from the intermediate mesoderm to form the metanephric mesenchyme. The metanephric mesenchyme forms the nephrons while the ureteric bud ramifies, forming the collecting tubules. Paired box protein 2 (PAX2) is known to be an important factor in kidney development; in metanephric mesenchyme, PAX2 is essential for sina-oculis-related homeobox 2 (Six2) gene activation. In vitro studies have shown PAX2 expression in renal vesicles, S-shaped bodies and distal renal tubules, confirming the necessity of the gene in kidney formation [5,9,10,11,12].

According to Jiang H et al., *PAX2* overexpression results in epithelial hyperproliferation and the occurrence of renal cysts [13]. *PAX2* downregulation restricts the mesenchymal to epithelial transition. In studies on mice, it was observed that the absence of the *PAX2* gene may stop renal organogenesis at incipient stages by significantly reducing the number of nephrons [14,15]. Other studies on mice have also shown the importance of *PAX2* in normal kidney development and the repercussions of *PAX2* overexpression, which is incompatible with normal kidney formation and results in kidney abnormalities that are comparable to human nephrotic syndromes [16]. Human glomerulogenesis can occur even in the absence of *PAX2*, although in vitro studies show the failure of the columnar to squamous transition in parietal epithelial cells in the absence of *PAX2* [12].

A heterozygous mutation (G189R) affecting the central octapeptide domain of PAX2, a crucial and conserved area, was discovered in a recent study [17]. Additionally, the study’s results support the idea that the G189R *PAX2* mutation has two effects that contribute to the adult onset of focal segmental glomerulosclerosis (FSGS): the most relevant one for the pathology’s onset occurs during the early stages of kidney development and may predispose the kidney to the development of FSGS; the second effect occurs in adult podocytes, which are less able to effectively defend off external triggers and are therefore more vulnerable to injury [17].

Congenital anomalies of the kidney and urinary tract (CAKUTs) cover a wide spectrum of lesions involving asymptomatic to severe, life-threatening malformations of the upper and lower urinary tract [18]. There is a high prevalence of CAKUTs in children, up to one per five hundred live births, with birth defects most frequently occurring in newborns [19]. Additionally, CAKUTs are the underlying cause of chronic kidney disease (CKD) in children, requiring renal replacement therapy in nearly 50% of cases [20,21]. About 25% of CAKUT cases have a genetic etiology [22].

CAKUTs can be identified as phenotypic components in hundreds of rare multi-organic syndromes. Ongoing CAKUT research has led to the discovery of novel CAKUT genes each year over the past 5 years, further expanding our already significant knowledge. These genes include those involved in interactions between transcription factors and epigenetic silencing complexes, as well as genes responsible for producing base membrane proteins and extracellular matrix proteins, which play a vital role in urinary branching by providing structural support and serving as a source of important growth factors. Additionally, CAKUTs have also been associated with various signaling pathways, some of which have also been linked to the development of malignant tumors [23].

Considering the *PAX2* gene is involved in the development of the kidney as well as the urogenital tract, eyes, ears and midbrain, we aimed to review described cases of renal disorders, without any ocular modifications, associated with the *PAX2* mutation, and to show the importance of this gene in kidney development and the anomalies that occur in patients with variants of this gene.

## 2. Literature Review for *PAX2*-Related Kidney Disorders

We carried out a literature review using the terms “*PAX2*”, “renal anomalies”, “nephrotic syndrome”, “focal segmental glomerulosclerosis”, “IgA nephropathy”, “cystic disorders” and “CAKUT” in PubMed, the MEDLINE database and Google Scholar. This research included an analysis of reported *PAX2* mutations, case reports of kidney disorders linked to *PAX2* and studies looking into the prevalence of *PAX2* genes in populations with kidney disease. Based on the genetic and phenotypic data, a meticulous analysis was performed.

Cases or studies lacking properly documented kidney involvement, those with eye involvement, or those without recognized pathogenic or likely pathogenic *PAX2* mutations were eliminated.

The inclusion criteria were as follows: cohort studies, case-control studies, narrative reviews, systematic reviews, human studies, *PAX2*-mutation-associated kidney disorders, and *PAX2* in pediatric patients.

The exclusion criteria were as follows: meta-analyses, non-renal *PAX2* mutations, and the *PAX2* mutation in adult populations.

Throughout the nephron’s maturation process, PAX2 is progressively downregulated starting from the podocyte progenitor cells of the glomerulus and then gradually in all of the proximal and distal tubules of the nephron. While the expression of PAX8 protein is stable in all epithelial cells of the mature nephron, the *PAX2* gene expression becomes limited to the collecting ducts and renal papilla [4].

Compound *PAX2/PAX8* heterozygous mutations in humans cause the near complete absence of the intermediate mesoderm with the absence of nephric ducts. These types of mutations also affect the branching morphogenesis of the developing kidney and interfere with the expression of LIM-homeobox 1 (LIM1), another transcription factor that plays a critical role in proper renal development. Since all members of the PAX family show haploinsufficiency, heterozygosity in both mice and humans leads to the emergence of symptoms due to the reduction in gene dosage [4].

The human kidney and urinary tract are highly dependent on gene dosage in the developmental phases. An important aspect of this transcription factor gene family is represented by the conserved DNA-binding paired box domain. *PAX2* is considered to be a target of transcriptional suppression by the *WT1* gene (Wilms tumor gene, a tumor suppressor gene). *PAX2* also influences the glial cell line-derived neurotrophic factor (GDNF) signaling, which partly mediates the correct insertion of the distal ureters in the bladder [4].

## 3. *PAX2* Gene and Kidneys

### 3.1. PAX2 Gene Mutations

*PAX2* heterozygous mutations were described for the first time in 1995 in patients presenting renal coloboma syndrome. In the following decade, most *PAX2*-related disorders were associated with renal coloboma syndrome. Lately, substantially more reported mutations in *PAX2* have been found in renal disorders exclusively [5].

Patients with the *PAX2* mutation associated with renal hypoplasia will develop end-stage renal disease (ESRD) in almost 100% of cases; therefore, patients with severe kidney malformations associated with ESRD should be tested for mutations of this gene [24].

Variants in the *PAX2* gene may lead to gene deficiencies, resulting in severe defects in the development of the urogenital system, namely CAKUTs [25].

Based on recent publications, Chang Y et al. found that the most frequent renal anomaly described is hypodysplasia; in cases of this, ESRD will develop at a median age of 10 years with a substantial degree of variability. Frameshift mutations were the most prevalent type of mutation observed, affecting nearly 50% of the individuals. Despite this finding, numerous patients with the same variants exhibit significant phenotypic heterogeneity [26].

Renal hypoplasia is a condition that consists of a reduction in the renal mass. So far, three types of renal hypoplasia have been described: simple hypoplasia, oligomeganephronia and segmental hypoplasia, which are similar to one another as they reduce the number of renal lobes. A fourth type has also been described which includes a medullary reduction and cortical thinning [27].

A different study that was conducted on 29 Brazilian kindreds who had the same *PAX2* mutation confirmed that renal hypoplasia was the most prevalent congenital defect in these patients [28]. The same study was continued in mice to determine in what manner the *PAX2* mutation causes renal hypoplasia [28]. The results suggested that *PAX2* haploinsufficiency may directly lead to renal hypoplasia. PAX2 reductions compromise the survival of the cells derived from the ureteric buds associated with the decrease in the branching of the ureteral bud in the incipient stage of kidney development. Additionally, PAX2, PAX5 and PAX8 also inhibit the transcription of p53, which results in cell apoptosis [28].

Because of the PAX2 protein’s influence over other key regulating factors in kidney development and regeneration, it is not completely understood how inhibiting this protein’s function would influence the complex cascade of regulatory genes and protein–protein interactions involved in CAKUTs and other renal and extra-renal pathologies (Figure 2).

### 3.2. PAX2 Mutations’ Consequences

The significantly variable genotype–phenotype association and the involvement of a multitude of different disease pathways were described in CAKUTs [21]. CAKUTs are a group of disorders comprising malformations of the kidney, such as renal hypoplasia, unilateral or bilateral renal agenesis (BRA), dysplasia, double kidney, polycystic kidney diseases (PKD), and anomalies of the ureter and bladder (vesicoureteral reflux (VUR), ureteropelvic junction obstructions) as well as the urethra (posterior urethral valves) [24].

Recently, Yamamura Y et al. identified three corresponding genes that are regulated by *PAX2* during human kidney development, namely *PBX1*, *POSTN* and *ITGA9* [31].

*PAX2* is one of the primary genes that is screened for mutations in cases with CAKUTs. According to previous studies, *PAX2* and *HNF1*-homeobox B (HNF1b) mutations can account for up to 15% of CAKUT cases. While mutations of *HNF1b* are typically described with cystic lesions of the kidney, *PAX2* mutations are more frequently associated with renal hypodysplasia [1].

*PAX2* gene mutations were reported to be associated with isolated kidney diseases, such as renal hypoplasia or dysplasia, in cases with kidney and urinary tract malformations and no ocular abnormalities [32].

### 3.3. PAX2 Mutations and Their Role in Children with Renal Pathology

#### 3.3.1. *PAX2* Mutations in Children with CAKUTs

A total of 349 published *PAX2* variants have been described so far. Of these, approximately 65% have been classified as pathogenic or likely pathogenic, and 90% of the variants are localized in the protein-coding regions [2].

In their study, Deng H et al. describe the frequency of the *PAX2* gene hotspot mutations that occur in the paired domain encoded in exons 2 and 4. They also emphasize phenotypic heterogeneity and clinical variability in children [33].

Until 2021, 92% of the *PAX2* mutations that had been described had kidney dysfunction of varied degrees, 77% presented colobomas or ocular dysplasia and 7% had hearing loss [34,35].

*PAX2* gene mutations in humans and mouse models have been reported to be associated with urinary tract and kidney anomalies.

According to Yang X et al., as of 2021, from a total of 234 patients described in the literature with kidney disease, 90 had been found to have pathogenic variants in *PAX2.* A total of 63% of those were identified with renal coloboma syndrome (RCS), 18% with CAKUTs, 8.4% with nephrotic syndrome, 0.9% with CAKUT and nephrotic syndrome, and 10.3% were patients with chronic kidney disease with an unknown etiology. The study points out the high prevalence of *PAX2* mutations; 38.4% of the patients were found to have variants in the *PAX2* gene [36].

A recent study that investigated *PAX2* gene mutations in 457 Japanese patients diagnosed with CAKUTs, cystic kidneys or renal dysfunction of unknown origin with or without any ocular abnormalities observed the presence of pathogenic *PAX2* variants in 38 cases (6.5%). The team identified two patients who had congenital cystic adenomatoid malformation (CCAM), which had never been reported before in association with *PAX2*-related disorders. It is possible that the source of this phenotype is completely unrelated to the *PAX2* mutation because previous in vitro studies suggested that the inhibition of PAX2 disrupts nephrogenesis without affecting lung development [37].

c.76_77insG is the most prevalent *PAX2* mutation. The same mutation was found by Zu R et al. in twins with FSGS [38]. In the ensuing decades, there have just been a few case reports in Western European and Japanese populations. The phenotype cannot be predicted by the point mutations. The phenotypes ranged from conventional RCS, which is characterized by a renal pathology associated with ocular defects, to CAKUTs, which presented as FSGS or renal hypoplasia. According to Barua M et al., 4% of adult FSGS cases are caused by *PAX2* mutations [39].

In a recent study involving 53 patients with syndromic and non-syndromic CAKUTs, Negrisolo S et al. described two novel variants [2]. One of the described variants was found in monozygotic twins. Both of the twins had renal cystic hypodysplasia, but only one of them was found to involve other organs, such as colobomatous cysts and congenital cataracts [2]. This is the second report of monozygotic twins carrying the same *PAX2* mutation with a discordant phenotype and different clinical course, pointing out that different RCS phenotypes can emerge from the same genotype [2,40]. The author’s conclusion was that there is an extreme level of phenotypic variability in papillorenal syndrome depending on the involved factors, such as epigenetics, environmental factors and modifying genes [2,40].

A recent genotype–phenotype analysis of patients with *PAX2* mutations concluded that 81.8% of patients with truncating mutations presented RCS, and 80% of patients with a missense mutation had FSGS or an isolated CAKUT, suggesting a strong association between the mutation locus and the presenting phenotype.

Additionally, they were able to identify that patients with mutations in the paired box domain showed a more aggressive progression of the disease than patients with mutations in other sites of the gene. However, the limited number of patients (*n* = 27) involved in this study, coupled with the relatively homogeneous ethnic background (all patients were South Korean), makes it unsuitable for drawing general conclusions [41].

#### 3.3.2. *PAX2* in Children with Other Renal Disorders

Numerous structural and functional kidney abnormalities can result from human mutations in the *PAX2* gene (Figure 3).

##### Renal Interstitial Fibrosis

Zhou T et al. highlight the negative correlation between Prohibitin (PHB) and *PAX2* in renal interstitial fibrosis (RIF) in a study conducted on rats. When PHB was less expressed, the *PAX2* gene expression increased. They also observed that *PAX2* overexpression contributed to RIF development. The researchers hypothesized that PHB might have a beneficial role in RIF by downregulating *PAX2* gene expression [42].

##### Renal Hypoplasia

In a study involving 10 patients with ages ranging from 2 to 15 years, Deng H et al. analyzed the variability in phenotype manifestation in children with *PAX2* gene mutations. All of the enrolled patients had an abnormal kidney structure and renal hypoplasia, and all were found to have proteinuria and renal dysfunction. Five of the patients had renal cysts, and three of them progressed to CKD. Ocular manifestations were absent in three of the patients, and only kidney injuries were present [33].

In the same context, a variability in renal phenotype was observed in a family involving three members with the same heterozygous *PAX2* mutation. One of them had isolated renal hypoplasia and ESRD, while the other two members had ocular manifestations, one of which had steroid-resistant focal segmental glomerulosclerosis and the other had renal hypoplasia [43].

In 2001, Nishimoto K et al. examined 20 patients with bilateral hypoplasia with decreased renal function; two of them were found to have a *PAX2* mutation, and both of them were born at term and had poor weight gain in the first month after birth. One of the two patients had an optic nerve coloboma, but the other had normal eye development. Until that moment, all of the described cases with *PAX2* mutations presented both renal and eye modifications. Therefore, this was the first described case of isolated renal hypoplasia [44].

Zhang L et al. presented a patient with renal atrophy and a *PAX2* mutation inherited from his father, who was diagnosed with kidney failure at the age of 20. The patient’s grandparents were tested as well, and both had a normal genotype and renal function. Based on that finding, the father’s mutation was considered as a de novo mutation. The described mutation, in this case, was a missense mutation. Additionally, non-sense and frame-shift mutations were described [14] with no correlation between any of the changes in the patient’s phenotype.

The deletion of the entire *PAX2* gene resulting in complete haploinsufficiency was first described in a case of a hypoplastic kidney without optic coloboma [45]. On the other hand, optic colobomas without renal anomalies in *PAX2* mutations were not detected [32].

In Salomon R’s study of the nine examined patients with isolated oligomeganephronia, a heterozygous mutation in the *PAX2* gene was found in three of them [46].

Rasmussen M et al. described a deceased fetus presenting with bilateral kidney agenesis. There was a considerable history of renal disease on the father’s side of the family, which was described by a novel pathogenic variant in the *PAX2* gene. Variable phenotypes were seen in the family history, including bilateral kidney hypoplasia, the obstruction of the ureteropelvic junction, hydronephrosis and duplex kidney [47].

##### Cystic Disorders

Fletcher J et al. describe a family with a heterozygous mutation in exon 2 of the *PAX2* gene; the missense mutation was present in six members of the family. All of the members had RCS with variable renal symptomatology, from medullary sponge kidney to bilateral renal hypoplasia, bilateral small dysplastic kidney and multicystic dysplastic kidney (MCDK) in association with optic disc coloboma [8].

A *PAX2* mutation was identified in a patient with a prenatal diagnosis of the Potter sequence, and a postnatal ultrasound revealed renal cysts [34]. The variant found in this case was not found in either of the patient’s parents, so it was considered a de novo mutation. The patient’s hearing and vision tests were normal [34].

In 2012, the first case of monozygotic twins with the same identified variant and a discordant phenotype was recorded [40]. One of the twins presented isolated renal anomalies, including multicystic kidneys, followed by severe renal insufficiency at the age of 20, without any eye abnormalities. On the other hand, the other twin had entirely lost one eye’s visual acuity by the time he was 2 years old, and, at the age of 20, he also had kidney disease, which was manifested by proteinuria, the same as the other twin [40].

##### Nephrotic Syndrome

The study conducted by Vivante A et al. demonstrated a high frequency (5.2%) of *PAX2* mutations in the familial form of steroid-resistant nephrotic syndrome (SRNS) [48]. Patients with other known monogenic causes of SRNS were excluded from this study. The team also identified a patient with associated microcephaly who carried a unique truncating mutation of the *PAX2* gene (c.69–70InsG; p. Val26Glyfs*28). This mutation was previously associated with brain development anomalies in a mouse model, suggesting that this phenotype could also be a result of this mutation and further expanding the phenotypic spectrum of *PAX2* heterozygous mutations to include autosomal dominant childhood-onset focal segmental glomerulosclerosis [48].

A 7-year-old child with nephrotic range proteinuria was found to have *PAX2*-associated nephropathy, according to a recent study [49]. The genetic and renal biopsy results were inconsistent. In contrast to the genetic diagnosis of FSGS type 7 caused by a mutation in the *PAX2* gene inherited from her father, the findings of the biopsy showed IgA nephropathy [49]. The molecular diagnosis altered how the biopsy results were interpreted in the cases that were described [49].

A cohort of 32 patients with *PAX2* nephropathy was described by Yang X et al. in 2021, with one patient having kidney biopsy results consistent with IgA nephropathy [36].

Moreover, Mansilla M et al. reported 54 cases with a genetic diagnosis in a cohort of 127 patients with kidney disorders. Out of these cases, four of them presented *PAX2* mutations with FSGS renal histopathology: a 15-year-old boy with FSGS (AD)/multicystic dysplastic kidney (c.419G>T, p.Arg140Leu), a 2-year-old boy with RCS (c.69delC, pVal26CysfsX2), a 20-year-old female with ESRD due to FSGS (c.70_71insG, p.Val26Glyfx*28) and an 11-year-old boy with IgA nephropathy or FSGS (c.1178G>C, pArg393Pro) (Table 1) [50].

In a large pediatric Chinese cohort with nephrotic syndrome (637 cases), 32 patients had nephropathy caused by a genetic mutation. Two (12.5%) of the sixteen patients with idiopathic SRNS who advanced to ESRD had *PAX2* mutations, and these patients were older than 6 years. One of them showed renal FSGS histopathology [51].

A recent article reported concomitant identical heterozygous mutations in *PAX2* (c.491C>A, p.Thr164Asn) and *MYO1E* genes in a pair of twin girls with steroid-dependent nephrotic syndrome and a similar phenotype but no extrarenal signs [52]. They have been considered to be of unknown significance.

**Table 1 ijms-24-12737-t001:** Variants in *PAX2* gene described in pediatric patients with renal disorders.

*PAX2* Gene Mutation	Molecular Consequence	Clinical Significance ^1^	Renal/Urinary Malformation	Reference
c.43+5G>A	Intron variant/splice donor 5th base variant	LP	Focal segmental glomerulosclerosis 7	[53]
c.76dup (p.Val26fs)	Frameshift	P	Unspecified cystic kidney disease	[54]
c.76dupG,(p.Val26Glyfs*28)	Frameshift	P	Focal segmental glomerulosclerosis 7/severe renal hypoplasia with end-stage renal disease	[43]
c.418C>G (p.Arg140Trp)	Missense	P	Renal hypoplasia	[14]
c.119–120del (p.Arg40fs)	Frameshift	P	Glomerulomegaly, Focal segmental glomerulosclerosis 7	[55]
c.187G>A (p.Gly63Ser)	Missense	P	Focal segmental glomerulosclerosis 7	[55]
c.576del (p.Ile193fs)	Frameshift	LP	Renal hypoplasia	[56]
c.68T>C (p.Leu23Pro)	Missense	P	Bilateral kidney agenesis	[44]
c.254G>T (p.Gly85Val)	Missense	LP	Focal segmental glomerulosclerosis 7, Steroid-resistant nephrotic syndrome	[40]
c.275C>T (p.Thr92Met)	Missense	LP	Focal segmental glomerulosclerosis 7, Steroid-resistant nephrotic syndrome	[40]
c.576del (p.Ile193fs)	Frameshift	LP	Renal hypoplasia	[56]
c.153_155delCTGinsTT	Frameshift	LP	Renal cystic hypodysplasia	[2]
c.155G>A	Frameshift	LP	Dysplastic multicystic kidney	[40]
c.69delC	Frameshift	P	Bilateral renal hypodysplasia, Vesicoureteral reflux and hearing loss	[2]
c.419G>T (p.Arg140Leu)	Missense	LP	Focal segmental glomerulosclerosis 7	[50]
c.69delC (p.Val26CysfsX2)	Frameshift	P	ESRD due to Focal segmental glomerulosclerosis 7	[50]
c.1178G>C (p.Arg393Pro)	Missense	P	Focal segmental glomerulosclerosis 7	[50]

^1^ Clinvar database https://www.ncbi.nlm.nih.gov/clinvar (accessed on 11 June 2023) and literature searching.

##### Urogenital Cancers

PAX2 is expressed aberrantly in different types of malignant tumors, such as some forms of ovarian cancer, RCC, Wilms tumors, prostate carcinomas and bladder carcinomas. Rare cases of pediatric RCC have been reported, mostly in teenagers and young adults. A high expression of the *PAX2* and *PAX8* genes is present in around 95% of RCC patients. Better clinical outcomes and the overall survival in RCC patients were linked to higher expression levels of PAX2 and PAX8 (PAX ClusterA expression subgroup) [57].

According to Sefidbakht S et al., nearly all of the Wilms tumor cases (91%) had PAX2 expression, while 82% of the cases had PAX8 expression. While the PAX2 and PAX8 expression did not differ substantially between tumors with favorable and unfavorable histology, larger tumors were linked with PAX8 expression [58].

In adult kidneys, PAX2 expression is limited to the medullary region as well as after acute kidney injuries, with some evidence suggesting that this re-expression might be involved in sustaining pluripotency and stem cell populations [59]. It was proven recently that PAX2 expression promotes angiogenesis in ovarian and colorectal carcinoma cell lines and also the formation of liver metastasis in colorectal carcinoma, suggesting its important role in oncogenesis [60,61,62].

## 4. Future Directions

Almost thirty years ago, scientists proved that the inactivation of *PAX2* by different approaches, such as antisense oligonucleotides, small interfering RNA (siRNA) or small hairpin RNA (shRNA), results in the growth inhibition of human renal cell carcinoma in vitro. Furthermore, it was confirmed that this approach increases apoptosis and sensitizes renal cancer cells to other chemotherapeutic drugs, suggesting that PAX2 can be an excellent target for therapeutic intervention in renal disease [63,64,65].

More recently, in silico and drug repurposing studies identified a small molecule, EG1, which is a highly effective and specific PAX2 inhibitor.

By attaching to this protein’s DNA binding site, EG1 can phenocopy a *PAX2* mutation to prevent PAX2–DNA interactions in vitro and in ex vivo kidney organ cultures; consequently, this could be beneficial in the future for the treatment of many urogenital cancers and renal malignancies. For this reason, further studies are needed to evaluate the safety and efficacy of this novel therapeutic approach in vivo [66].

Bradford S et al. reported the identification of small-molecule inhibitors for the *PAX2* family of developmental regulatory proteins. Such inhibitors can suppress the proliferation of PAX2-expressing cancer cells and may provide scaffolds for future anti-cancer compounds [67].

From what we have learned so far from the literature, the majority of the cases with mutations in *PAX2* were diagnosed with RCS; therefore, we had a smaller number of patients included in this article, considering the fact that we included only patients with renal anomalies.

Finding *PAX2* gene variants is crucial because they may have an impact on how patients are treated. Glucocorticoids or immunosuppressants should not be provided to patients with genetic FSGS or other genetic nephropathies because they have no effect in these circumstances. Consequently, serious side effects could be avoided.

Our review focuses on pediatric renal disorders associated with *PAX2* mutations. To the best of our knowledge, this is the first review that includes patients who have been identified with *PAX2* mutations associated exclusively renal abnormalities such as CAKUTs, renal interstitial fibrosis, renal hypoplasia, cystic disorders, nephrotic syndrome and urogenital cancers. In this article, we focus on the associated genetic mutation, the genotype–phenotype association and the clinical presentations of the involved patients.

Considering that *PAX2* mutation frequency seems to be underestimated when treating patients who have CAKUTs, chronic kidney illnesses with no known cause, the involvement of several systems, and/or family histories of renal disease, clinicians need to be vigilant and look for the *PAX2* mutation. According to the fact that patients with the *PAX2* mutation associated with renal hypoplasia will develop ESRD in almost 100% of cases, it is crucial for all children presenting severe kidney malformations associated with ESRD to be tested for this mutation. A genetic cause should be investigated in ESRD cases of unknown etiology as the above-mentioned patients will require kidney transplantation. The identification of these patients with the *PAX2* mutation is essential to optimizing pre- and post-kidney transplantation management. Clinicians and family members must make important medical and personal decisions based on an early and precise genetic diagnosis.

In conclusion, *PAX2* plays a crucial role in kidney organogenesis, and variants in this gene lead to various types of kidney anomalies and disorders. The present work summarizes pediatric patients found to have mutations in the *PAX2* gene related to kidney disease without the involvement of other organs. Because of the high prevalence of renal diseases, especially CAKUTs, in pediatric patients and their significant impact on patients’ quality of life and lifespan, we point out that patients with kidney disorders of unknown cause should be tested for *PAX2* mutations.

## Figures and Tables

**Figure 2 ijms-24-12737-f002:**
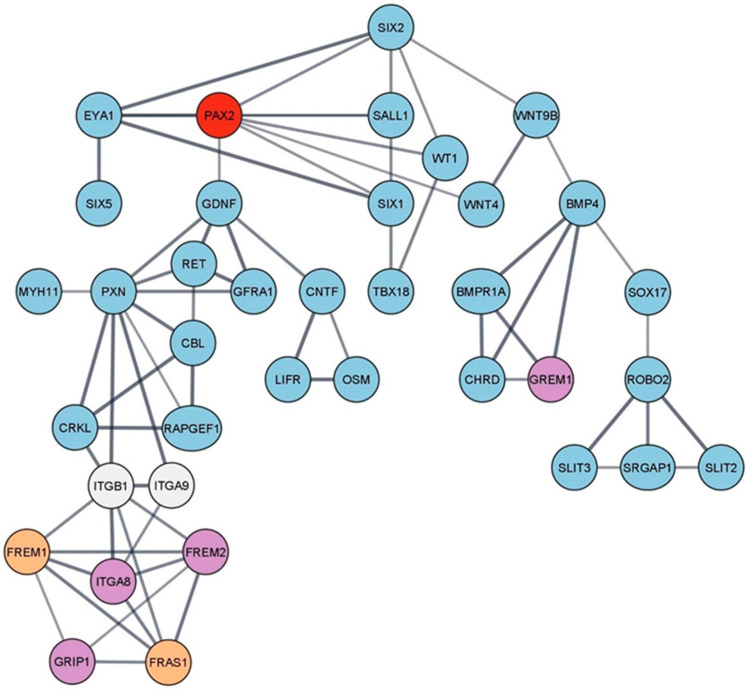
Visual representation of protein–protein associations between major proteins that are encoded by CAKUTs and syndromic CAKUT-causing genes. The main network was constructed using the STRING database and processed in Cytoscape. A minimum interaction score of 0.7 (high confidence) was used. Our protein of interest, PAX2, is shown as a red node. Blue nodes represent proteins coded by CAKUT-causing genes. The purple nodes are proteins that are encoded by syndromic CAKUT-causing genes. Orange nodes represent an overlap between the two categories. The white nodes were manually added to the network. Nodes that were not connected to the network were eliminated. The edges indicate both functional and physical protein associations, while the thickness of the edges is proportional to the strength of the data support [23,24,29,30].

**Figure 3 ijms-24-12737-f003:**
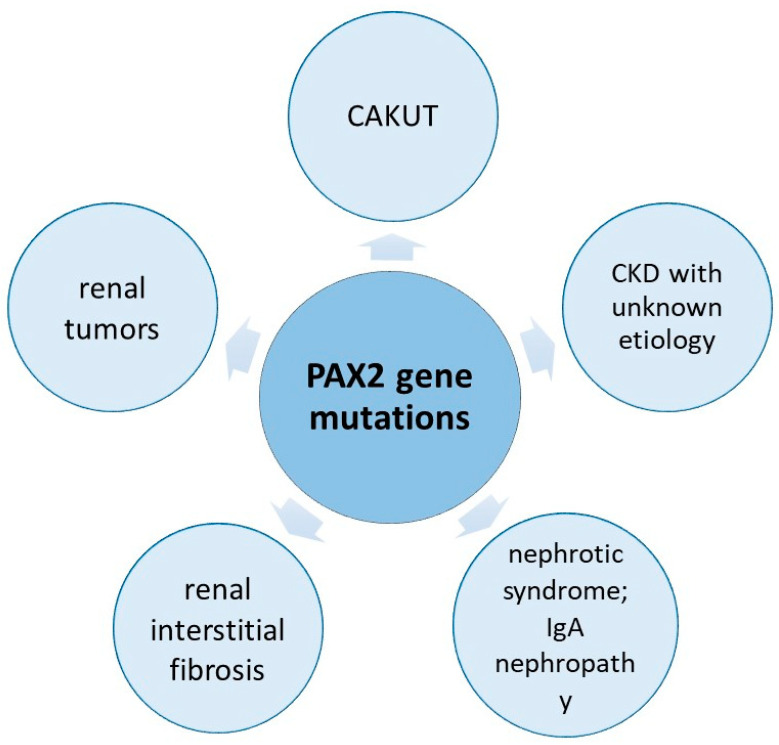
Visual representation of the phenotypic heterogeneity of *PAX2* mutations associated with kidney disease.

## Data Availability

Not applicable.

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
