# Peer review of "PAX2 Gene Mutation in Pediatric Renal Disorders—A Narrative Review"

_ijms, 2023, doi:10.3390/ijms241612737_

Round 1

Reviewer 1 Report

It is recommended to clearly define the scope of the review, such as the specific types of pediatric renal disorders associated with PAX2 mutations that will be discussed. Additionally, it would be beneficial to clarify whether the review will focus on the clinical presentation, genetic testing methods, treatment options, or a combination of these aspects.

The review should include the most recent studies and findings related to the PAX2 gene mutation in pediatric renal disorders. This will ensure that the review reflects the current state of knowledge in the field.

The review should be organized and structured in a logical manner. Consider dividing the content into sections or subsections based on different aspects of PAX2-related renal disorders to improve readability and comprehension. Each section should have a clear heading that reflects its content.

As this is a narrative review, it would be beneficial to describe the methodology used for selecting and analyzing the literature. For example, specify the databases searched, inclusion/exclusion criteria for studies, and any specific analysis methods employed.

While discussing the PAX2 gene mutation, it is important to address the clinical relevance of these findings. How does the presence of a PAX2 mutation impact the diagnosis, prognosis, and treatment of pediatric renal disorders? Are there any specific challenges or considerations that arise when managing patients with PAX2 mutations?

Consider including a section on future directions for research or clinical practice related to PAX2 gene mutations in pediatric renal disorders. This can help identify gaps in knowledge and potential areas for further investigation.

Minor editing of English language required

Reviewer 2 Report

The submitted manuscript addresses PAX2 gene mutations in pediatric renal disorders. Please find my comments below.

1.  Line 39 - his role?

2. Line 40-41 - Please modify the sentence so that it is clear.

3. All references, for example [1], are after period/full stop. Is it correct?

4.  Figure 1 - pls check for grammatical errors. Is it correct to use future tense in figure legend? will in line 64?

5. line 71-75 - length sentence. please modify it.

6. line 79 - paraphrase the sentence.

7. Many of the sentences in this article are very lengthy and not easily understandable. Thorough proof-reading is required.

8. Inclusion of schematic on pax2 mutations and associated disease conditions is much appreciated.

9.  Contents of Table 1 have different fonts. 

10.  Why main focus on CAKUT, no other specific renal abnormalities?

11. The flow of this article is not good. 

Extensive editing and proof-reading are needed. 

Round 2

Reviewer 2 Report

Thanks for revising the manuscript. Please find my comments. 

1. Line 202 - Typo error 'Concequences'?

2. Line 223 - 65% are classified - correct the typo!

3. Please change/replace the images as they are blurred, for example figure 3. Figure 1 should be replaced with better one.

The manuscript still has typo errors. Therefore, it is strongly recommended to revise it. In some places, you can find 2-3 sentences, not a typical paragraph style (line 322-333; 415-430).

Round 3

Reviewer 2 Report

Thanks for revising the manuscript. The figures are still blurred. Can you change it if possible? 
